# Basic Theory of Ice Crystallization Based on Water Molecular Structure and Ice Structure

**DOI:** 10.3390/foods13172773

**Published:** 2024-08-30

**Authors:** Ouyang Zheng, Li Zhang, Qinxiu Sun, Shucheng Liu

**Affiliations:** 1College of Food Science and Technology, Guangdong Ocean University, Guangdong Provincial Key Laboratory of Aquatic Product Processing and Safety, Guangdong Province Engineering Laboratory for Marine Biological Products, Guangdong Provincial Engineering Technology Research Center of Seafood, Key Laboratory of Advanced Processing of Aquatic Product of Guangdong Higher Education Institution, Zhanjiang 524088, China; zhengouyang07@163.com (O.Z.); 13822507024@163.com (L.Z.); lsc771017@163.com (S.L.); 2Collaborative Innovation Center of Seafood Deep Processing, Dalian Polytechnic University, Dalian 116034, China

**Keywords:** frozen food, ice crystallization, water and ice crystal structure, ice crystallization characterization

## Abstract

Freezing storage is the most common method of food preservation and the formation of ice crystals during freezing has an important impact on food quality. The water molecular structure, mechanism of ice crystal formation, and ice crystal structure are elaborated in the present review. Meanwhile the methods of ice crystal characterization are outlined. It is concluded that the distribution of the water molecule cluster structure during the crystallization process directly affects the formed ice crystals’ structure, but the intrinsic relationship needs to be further investigated. The morphology and distribution of ice crystals can be observed by experimental methods while simulation methods provide the possibility to study the molecular structure changes in water and ice crystals. It is hoped that this review will provide more information about ice crystallization and promote the control of ice crystals in frozen foods.

## 1. Introduction

Food is highly susceptible to spoilage because of the endogenous enzymes, microorganisms, and other physiological and biochemical reactions. Freezing can extend the shelf-life of food at subzero temperatures by removing large amounts of water from the solute phase to inhibit chemical, microbial, physical, and other reactions [1]; therefore, freezing preservation is the most common method of food preservation. However, the volume increase caused by the phase transition of water during freezing may irreversibly disrupt the cell structure, which directly affect the quality, economic value, and consumer acceptance of food products [2]. In addition, the crystallization and recrystallization of ice will also cause physical and chemical changes in food components such as protein freezing denaturation and lipid oxidation, which have negative effects on nutrition, texture, color, and flavor [3]. In order to inhibit freezing damage, it is of great importance to accurately predict and control the freezing process of water under various conditions. The size, shape, and distribution of ice crystals during freezing determine the degree of mechanical damage to food tissue, which depends on factors such as the nucleation temperature, freezing point, temperature fluctuations, moisture state, supercooling, and freezing rate [4].

Water molecules are connected to each other by hydrogen bonding to form different water molecule clusters. Changes in the structure of water molecular clusters lead to changes in water properties such as viscosity, enthalpy, and surface tension [5], which, in turn, affect the water state, supercooling, and the freezing rate, therefore changing the morphology of the final ice crystal formation. In frozen tissue foods, the formation of large ice crystals outside the cell leads to various harmful effects on the tissue including mechanical damage, low-temperature concentration, frozen burns, and recrystallization. These phenomena may occur in metabolic system disorders, enzyme system misalignment, and cell membrane damage and ultimately lead to cell lysis. On the other hand, the formation of uniformly distributed fine crystals inside and outside the cell can better preserve the quality of the product to the tissue due to minimal damage [6]. For ice cream products, the number and sizes of ice crystals directly determine whether the taste of ice cream is smooth or rough. Large ice crystals can disrupt the smooth texture of ice cream. Not only that, ice crystals also determine the skeleton of ice cream. However, crystals are more needed in processes such as freeze drying and freeze concentration [7]. Therefore, controlling, understanding, and predicting the ice crystallization process and related phenomena is crucial for improving food freezing technology.

In order to study the formation of ice crystals in food, direct and indirect methods of observing ice crystals have been continuously explored and developed [8]. As imaging and spectroscopic techniques continue to advance, the information obtained provides a more reliable understanding of the freezing process. With the enhancement of computer capabilities, simulations have become a powerful tool for studying the mechanism of the ice crystal growth process [9]. The use of computer simulations (e.g., cellular automata method, molecular dynamics, phase-field method, and Monte Carlo) to study the growth mechanism of ice crystals has largely contributed to the understanding of crystallization at the molecular scale [10]. Based on the understanding of ice crystal formation, various techniques have been proposed to control ice crystal formation, including the use of physical fields, liquid gas, and exogenous additives.

With the increasing interest in crystallization, there have been many reviews on crystallization detection methods and crystallization control techniques [11,12], but there have been fewer systematic reviews on the water molecular cluster structure, ice crystal structure, ice crystal characterization methods, and ice crystal control techniques. In fact, the in-depth understanding of the water molecular cluster structure and ice crystal structure will be more beneficial to develop and reveal the mechanism of new freezing technology; therefore, the systematic elaboration of the water molecular cluster structure and ice crystal structure and ice crystal characterization are necessary to control ice crystal behavior and improve the quality of frozen food. Therefore, this review intends to systematically review the ice crystallization process from the aspects of the ice crystal formation mechanism, water molecule and ice crystal structures, and ice crystal characterization methods. It is hoped to provide information for people to control the dynamic process from water to ice.

## 2. Structure of Water Molecular Cluster

Water has a relatively simple geometry: two hydrogens and one oxygen, H_2_O, form a tetrahedral structure [13]. However, such a simple structure induces complex interactions, making it difficult to predict water behavior [14]. In the water molecule, there are 10 electrons (five pairs) (Figure 1A); one pair of electrons (internal) is located near the oxygen nucleus. The remaining four pairs of electrons (external), one pair between the oxygen nucleus and each hydrogen nucleus, and the remaining two pairs, are lone pairs of electrons that play an important role in the generation of hydrogen bonds between molecules. Two bonding pairs of electrons and two lone pairs of electrons are distributed in a tetrahedral pattern on four vertices. This tetrahedral configuration of the water molecule and its ability to form hydrogen bonds with other molecules from four directions is a major factor in the formation of hydrogen bonds. The ability to form hydrogen bonds with other molecules from four directions is the basis for understanding the structure and properties of water in all water-containing compounds. The bond angle between two covalent bonds O-H in liquid water is 105°. Oxygen atoms are more capable of attracting electrons than hydrogen atoms. The charge distribution of the water molecule is not uniform; each hydrogen atom is slightly positively charged whereas each oxygen atom is slightly negatively charged, which explains the water molecule’s dipole moment and considerable dipole polarization polarity (Figure 1B). Because of their polarity, water molecules can interact with one another and form a network by forming numerous intermolecular hydrogen bonds with their nearest neighbors. Each water molecule has the ability to form up to four hydrogen bonds. Oxygen may establish hydrogen bonds with two hydrogen atoms and each of the two hydrogens can form a hydrogen bond.

People have tried to determine the properties of water by all means, hoping to get information about its structure and propose a structural model. But in liquid water, the molecules are constantly in thermal motion and the relative positions between molecules are constantly changing. It is impossible for water to have a single, definite structure like a crystal. Theoretically, each water molecule can act as both a hydrogen bond acceptor (i.e., provide a lone pair of electrons) and a hydrogen atom to form a hydrogen bond (i.e., hydrogen bond donor), which can form a dynamic network of hydrogen bonds in liquid water, forming a water molecule chain, or a water molecule cluster structure. This cluster structure of water molecules is essentially composed of a large number of hydrogen bonding chains. The strength of a hydrogen bond depends mainly on the bond length and the bond angle between the hydrogen bond and the covalent bond. The bond length is the most important one and the strength of hydrogen bond decays exponentially as the bond length increases.

Two adjacent water molecules in water can form some dimeric water molecule clusters, trimeric water molecule clusters, and other multimerized water molecule clusters through hydrogen bonding or dipole-a-dipole interactions, thus forming a long chain of water molecule hydrogen bonds.

Dimeric water is the smallest unit of water molecule cluster, and this exists mainly in the gas phase, and its structure is linearly distributed with hydrogen bonds (Figure 1D). It is a cluster of water molecules formed by the polymerization of two water molecules (a hydrogen atom within one water molecule and an oxygen atom of another water molecule). The dimeric water can absorb the infrared light from the sun so that the lower atmosphere does not get the proper infrared radiation from the sun, thus causing the temperature to drop significantly [15]. According to research, in 30 °C and 70 °C conditions, this effect can make the ground cool by 4 °C.

There is another kind of water called polymerized water. This water can be obtained by injecting clean water vapor into a capillary tube, putting it into a sealed box, pumping out the air and heating it to 300 °C, keeping it for several hours, then passing it in ordinary water vapor and letting it cool [16]. This water is very viscous, like petroleum jelly, and condenses into a glassy body below −40 °C and is as thick as resin at −100 °C. It is very stable and can maintain this strange property up to about 500 °C. It can recover to normal water only when heated up to 700 °C. Regarding the structure of this water, it is believed that under certain conditions, water molecules are arranged in an orderly manner, in groups of three or four molecules, very firmly polymerized together. These polymerizations are difficult to move and, as a result, are dense and viscous, and even at low temperatures, these molecules cease thermal movement and do not freeze into ice [16].

The trimeric water appears geometrically as an approximately planar triangle (Figure 1E). The role of water intermolecular hydrogen bonding in trimeric water mainly exhibits cooperativity, i.e., when the first hydrogen bond is formed, the charge distribution in a monomer molecule involved in the formation of a hydrogen bond will change and a hydrogen acceptor molecule will become a potential or even better hydrogen donor than before. In other words, a stronger second hydrogen bond is able to be formed due to the presence of the first hydrogen bond. Thus, water molecules have the propensity to associate into clusters and thereby compose a dynamic hydrogen bonding network.

The structures of tetrameric water and pentameric water are shown in Figure 1F,G. The equilibrium structure of tetrameric water is a quasi-planar structure of the S4 symmetric type with an oxygen–oxygen bond length of 2.74 Å [17]. Most of the pentameric water shows a quasi-planar in-ring structure similar to the cyclopentane bending shape. The O-O-O angle of this minimal polygon is close to the optimal tetrahedral angle value, thus forming the maximum hydrogen bond energy. Therefore, pentameric water is relatively stable and occupies a large proportion of the free water.

Hexameric water is a transition cluster from two-dimensional planar ring-shaped water molecule clusters to three-dimensional structured water molecule clusters (Figure 1H). Thus, hexameric water exhibits both planar and spatial structural features, mainly six-ring, book-shaped, ship-shaped, prismatic, and cage-shaped features.

In liquid water, molecules continuously undergo thermal motion and the relative positions between molecules constantly change. However, a large number of hydrogen bonds remain between molecules in water, connecting them together. In addition to the irregular distribution of ice structural fragments among molecules, there are also a large number of dynamically balanced, incomplete polyhedral connections. The main forms of polyhedrons are pentagonal dodecahedrons, as well as other polyhedrons composed of five-membered rings and six-membered rings.

Water association is a process in which the simple molecules contained in water combine to form complex molecular groups without causing a change in the chemical properties of the substance. It can be expressed as a reversible process by the reaction equation nH_2_O=(H_2_O)n. Association versus dissociation does not occur when an associating molecule is in equilibrium with a simple molecule, and once this reversible process is disrupted, it is the process of water becoming ice or ice becoming water or repeated cycles of ice versus water.

Association is an exothermic process and dissociation is an endothermic process; therefore, the temperature increases, the extent of water association decreases (n decreases), and at high temperatures, water mainly exists in a monomolecular state. The temperature decreases, the association of water increases (n increases), water turns into ice at 0 °C, and all water molecules associate into one giant molecule.

Ice is a crystal structure formed by the orderly arrangement of water molecules and the arrangement of water molecules in the crystal of ice is such that each oxygen atom has four hydrogen atoms as its close neighbors, which two hydrogen atoms and oxygen atoms are linked by electron pair bonds and two hydrogen atoms and oxygen are linked by hydrogen bonds to form a crystal structure in which each water molecule is tetrahedral (Figure 1C). Such an arrangement leads to the formation of an open structure; that is, there are considerable holes in the ice. Therefore, the density on ice is relatively small. The process of melting ice into water is the process of breaking partial hydrogen bonds (about 15%), forming ice “fragments”, and partially cyclizing into polyhedrons represented by pyritohedron (Figure 2).

In summary, water molecular cluster structures directly affect the water crystallization process, and therefore, a better understanding of water molecular structures will be beneficial for regulating ice crystal production in frozen foods.

## 3. Mechanism of Ice Formation

The formation of ice crystals is the most obvious and direct result of food freezing. This unavoidable mechanism is the main source of food damage caused by freezing [18]. Therefore, understanding the mechanism of ice formation is very important to improve the quality of frozen foods.

The ice crystal formation process consists mainly of crystallization (nucleation and subsequent growth of ice crystals) and recrystallization (caused by temperature fluctuations) [18], which are controlled by diffusion-limited growth and molecular surface attachment dynamics [19]. The freezing process involves three key steps: (i) pre-cooling, (ii) phase transformation, and (iii) tempering [20]. During the pre-cooling step (Figure 3A), sensible heat in the food is removed and the temperature is lowered. When the solution is cooled to its initial freezing point (*T*_f_), it does not freeze spontaneously. Due to the fluctuation of density in liquid water, water molecules organize into clusters with the same molecular configuration as ice crystals but remain liquid due to the fluctuation of energy, which is called supercooling (difference between *T*_f_ and *T*_n_) [5,21]. This is related to the nature of the solution, the cooling rate, the process conditions, and the roughness of the vessel surface. The surfaces, suspended particles, impurities, and tiny bubbles of heat exchangers and containers can promote heterogeneous nucleation and reduce superercooling. For pure supercooled liquids, they need to undergo spontaneous density or composition fluctuations and undergo homogeneous nucleation and the degree of supercooling is related to the density of the solution. The degree of supercooling also depends on the cooling rate. However, there has been no unified conclusion on the relationship between supercooling and the cooling rate. Some researchers believe that increasing the cooling rate can increase supercooling while others believe that increasing the cooling rate can reduce supercooling. In addition, researchers have also found that the higher the surface roughness of the container is, the lower the supercooling will be [22]. The maintenance of superior-quality frozen food may be achieved by increasing supercooling during the freezing process, which could result in a significant number of tiny ice crystals throughout the food system [23]. According to the classical theory of water freezing nucleation, for every 1 °C decrease in supercooling temperature, the nucleation rate increases geometrically [24]. Water molecules organize into clusters with relatively lengthy hydrogen bonding activity as a result of density fluctuations brought on by Brownian motion in supercooled materials, which is analogous to molecular rearrangements in ice crystals [25]. Since this process is energetically unfavorable, these clusters break down rapidly. Once the critical mass of the nucleus is reached (*T*_n_, nucleation temperature), ice crystallization occurs rapidly in the whole food matrix. The nucleation of ice crystals includes homogeneous and heterogeneous nucleation. In pure water, homogeneous nucleation usually occurs, which requires very large supercooling. In food systems, heterogeneous nucleation is the most common nucleation mechanism. Although supercooling is required to initiate nucleation in foods, it is usually much lower due to the presence of many solid particles, solutes, and surfaces used as nucleating agents [20]. Therefore, in complex food systems, the undercooling of food is often very low, which is even difficult to observe (Figure 3B). The structure and size distribution of the crystals are mostly determined by ice nucleation [26]. When crystallization starts, the temperature of the food rises to the equilibrium freezing point (*T*_f_) because the latent heat generated during ice formation is faster than the heat lost during further cooling. The temperature increase due to ice crystallization usually prevents any subsequent nucleation of the product center. Only after nucleation does crystal growth take place by adding water molecules to the already formed nuclei; the rate of growth is regulated by the effectiveness of latent heat removal and the rate of cooling [25]. The temperature decreases rapidly once the ice formation is nearly complete (Figure 3A, tempering step). The formation of ice crystals during freezing increases the concentration of solutes in the unfrozen phase of the food system, resulting in a significant increase in the viscosity of the frozen concentrate matrix [27]. In high-viscosity systems, there is a significant reduction in molecular mobility, which prevents the solutes from crystallizing [27].

When all of the water that can freeze is totally turned into ice and the ice production ends within a reasonable period of time, the freezing concentration has reached its maximum composition [29]. In general, 90% of the moisture in frozen food tissue is frozen into ice when the temperature drops below −10 °C. The water in frozen foods is non-freezable water, which means that it does not freeze even at very low temperatures due to its extremely high viscosity, which involves bound water (combined with macromolecules in the matrix) and partially uncrystallized free water [30].

Recrystallization, which is characterized by a drop in the ice crystal number and an increase in the ice crystal size, invariably takes place throughout the storage and transit of frozen foods [6]. The system approaches an equilibrium state where the free energy is minimized and the chemical potentials between all phases are equal, resulting in the fact that recrystallization occurs. Due to the high surface-to-volume ratio of small ice crystals, which are thermodynamically unstable, water molecules have a tendency to separate from them, then diffuse through the frozen concentrated solution and finally redeposit on the surfaces of the larger crystals. This process helps lower the system’s surface energy. As a result, all of the smaller crystals contract while the larger ones expand, increasing the average size overall [6]. Similarly, an ice crystal with a rough surface (higher surface/volume ratio) has a larger surface free energy, and in order to limit its energy level, it develops a more compact structure (rounding or smoothing the ice surface) with a smaller surface-to-volume ratio (lower surface free energy).

## 4. Structure of Ice Crystal

Ice crystals exist in a wide variety of forms, from elongated cylinders to thin plates, sometimes branched, scalloped, hollow, and faceted [31]. Ice crystals, as in (i) and (ii), are almost always generated in a narrow temperature range around −15 °C. Only when the temperature is close to −5 °C, slender columnar and acicular crystals (iii) will appear. When the growth conditions change with the development of a crystal, a composite snow crystal type, such as a capped column (iv), will be formed. However, after much experimental and theoretical research, it is still unknown why these distinct architectures develop under different growing settings. For example, despite the fact that this behavior was originally observed 70 years ago, we still do not have a qualitative understanding of why ice crystal development alternates between plate and columnar forms as a function of temperature [31].

Due to the presence of hydrogen bonds between water molecules, whose strength and site shape may be flexible altered, water–ice shows an exceptionally rich and complicated phase diagram [32]. At least 18 different ice crystal structures (Figure 4A) have been synthesized in the laboratory, including ice Ih, ice Ic, ice II, ice III, and ice XVII. The oxygen atom positions in these ice phases are fixed while the hydrogen atoms are somewhat ordered and some are disordered. At different temperatures and pressures, water molecules exhibit different hydrogen bonding networks; however, there are four close neighbors that are connected by hydrogen bonds to each water molecule. The O-O-O angle in low-pressure ice Ih is around 109°, which is the ideal tetrahedral angle [32]. The molecules must be reconfigured to take up less space as the pressure rises, initially by altering the network structure (while maintaining tetra-coordination) and increasing the deformation of the O-O-O angles. For example, in ice II, these angles vary between 80° and 129°. The available reduced volume can no longer be filled just by increased hydrogen bond distortion when the pressure rises further. The water molecules then come together to create an interpenetrating network, such to that of ice VII, which is mostly made up of two interpenetrating rhombic lattices (Figure 5). But each network still has four synchronized channels.

Most of the 18 kinds of ice crystals have been obtained by adjusting the pressure and temperature in an artificial experimental environment. At present, only hexagonal-phase ice (ice Ih) and cubic-phase ice (ice Ic) can be formed and exist spontaneously in nature. Among them, hexagonal-phase ice Ih was the first to be recognized and studied because it requires fewer conditions for formation, and it is the most widely distributed in nature. Ih is a relatively open low-density crystal (density about 0.926 g/cm^3^), a six-fold axially high symmetry with hexagonal ice, and its basic cell unit can be seen as a hexagonal box composed of two reversed six-membered rings. All snowflakes and ice formed on earth by water vapor crystallization are in the form of hexagonal ice crystals. Cubic ice Ic crystals have a metastable face-centered cubic structure [33] and are formed by the condensation of water vapor at a low temperature (below 80 °C), crystallization in water droplets at below 30 °C [34], or phase transition from high-density ice by decompression at 77 K [35]. Compared with hexagonal ice, the metastability of cubic ice is caused by its higher symmetry. Usually, with the passage of time, cubic ice will gradually evolve into hexagonal ice.

**Figure 5 foods-13-02773-f005:**
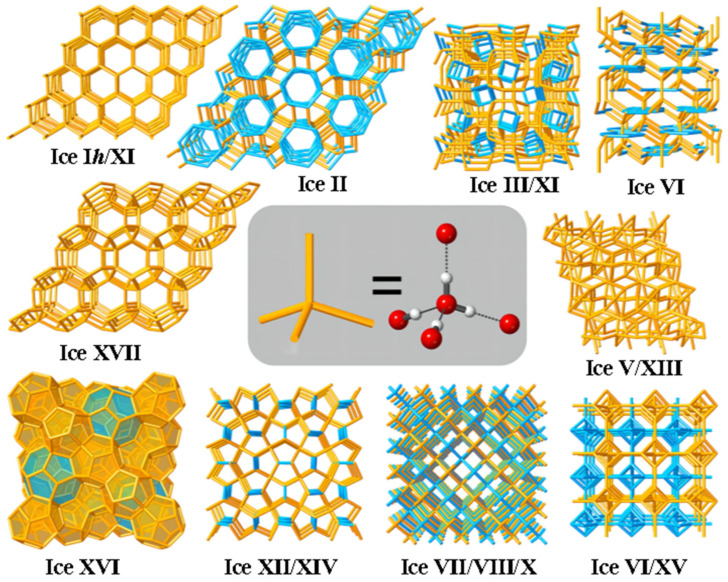
Hydrogen-bonded networks of the various polymorphs of ice. Reprinted with permission from Ref. [36]. Copyright 8 February 2019, copyright Salzmann, Christoph G.

Only five kinds of ice (Ih, III, V, VI, and VII) share the phase boundary with the liquid phase [32]. Ice III is a tetragonal crystal with a proton disorder, which is obtained by cooling liquid water to 250 K at 300 MPa and has a narrow region in the phase diagram. Ice V is a monoclinic crystal with proton disorder [37], Ice VI is tetragonal crystal with a hydrogen bond disorder [38]; ice VII is a cubic crystal with a proton disorder [39]. Ice Ih is the only ice type with lower density than water (0.92 g/cm^3^). The densities of ice III, V, VI, and VII are 1.14 g/cm^3^, 1.23 g/cm^3^, 1.31 g/cm^3^, and 1.591 g/cm^3^, respectively, and there is no volume expansion phenomenon during crystallization; that is, the mechanical damage to food tissues will be minimized. It is worth noting that the ice crystals in VI region cannot exist for a long time, and once formed, they will rapidly transform into ice II and III.

Using the density of hexagonal ice Ih or ice XI (about 0.93 g/cm^3^) as a reference, most of the ice phases have densities greater than this value, and only cage ice XVI (type II cage hydrate), cage ice XVII, cage ice I, and cage ice H from the hydrated phase have densities lower than this value. In conclusion, the number of known low-density ice phases is relatively small compared to the number of high-density phases. In the phase diagram of water ice, it can be seen that when the temperature is constant, the phase transition occurs continuously with increasing pressure, accompanied by an increase in the density of the ice phase. Conversely, if a series of ice phases with decreasing density is found, it should be possible to expand the phase diagram under negative pressure. Kosyakov and Shestakov [40] provided the temperature (T)–pressure (P) phase diagrams of liquid water (Figure 4B), hexagonal ice, and two hypothetical low-density I and II cage ice phases under negative pressure by estimating their Gibbs free energies. The triple points of liquid water, hexagonal ice, and type II cage ice were given in the phase diagrams, but type I cage ice did not appear in the phase diagrams because of its high free energy. Compared with hexagonal ice, cage ice II can exist stably in a lower-negative-pressure area, which shows that an ice phase with lower density can exist in lower-negative-pressure conditions. However, there are still great challenges in the experimental synthesis of the new negative-pressure ice phase structure predicted by theory, which needs to be further studied from both theoretical and experimental aspects.

The change in ice crystal volume during freezing and storage is an important factor causing freezing damage to foods, so understanding the conditions of ice crystal structure change is essential to reveal the mechanism of ice crystal formation controlled by new technologies and to control the formation of ice crystals in foods by rational use of new technologies. However, at present, ice crystal structure research is mostly focused on meteorology and physics while food freezing research focuses on freezing methods and food quality, and there is a lack of effective combination between the ice crystal structure formation mechanism and food tissue damage.

## 5. Characterization Methods for Ice Crystals

The effective characterization of ice crystal formation in food is the prerequisite for studying the influence of ice crystal formation on food tissue and developing new technologies to control ice crystal formation. With the development of science and technology, some experimental methods have been used to detect ice crystals, including light microscopy, electron microscopy, X-ray, differential scanning calorimetry, and some new imaging techniques (Table 1). Most of these methods are used for microscopic observation after ice crystal formation, but it is impossible to observe the changes in water molecular structure and the interaction between water molecules and other macromolecules during ice crystal formation. Therefore, some simulation techniques have been developed to study ice crystal formation during the freezing process including Monte Carlo simulation and molecular dynamic (MD) simulation.

### 5.1. Light Microscopy

Light microscopy is a commonly used method to indirectly observe the morphology and distribution of ice crystals in food tissues. Before observation, it is necessary to remove water from the sample [8] and then use a light microscope to observe the microstructure of frozen food at room temperature. Therefore, an initial preparation step (tissue treatment) is required to fix the original structure to the maximum level and avoid morphological deformation before being observed through a light microscope. According to different pre-treatment methods, light microscopy can provide microscale information about the structures of frozen food, freeze-dried food, and thawed samples [41].

The two most popular pre-treatment techniques for examining the distribution of frozen ice crystals using light microscopy are freeze substitution and freeze drying [6]. Fixation and substitution are the two stages in the freeze substitution process. The sample is submerged in a fixation solution that migrates into the sample and forms cross-links between and among the molecules of the tissue components, preserving the structure and maintaining the conformational connection of the ice/water phase [42] (Figure 6A). The ice crystals and water are eliminated during the replacement process using a polar organic solvent. The matrix can be observed directly after the organic solvent has evaporated or after embedding [42]. By using freezing substitution, the ice crystal size, shape, and distribution can be qualitatively assessed. In addition, the images can also be processed by some image analysis software (such as ImagePro, ImagePlus, etc.) to qualitatively characterize the characteristics of ice crystals including the fractal dimension, ice crystal diameter, ice crystal roundness, elongation, and ice crystal area.

The removal of moisture from frozen samples can also be achieved by the sublimation of ice crystals using the freeze drying method, and the subsequent steps are similar to the conventional procedure followed in freeze substitution. Samples prepared by the freeze drying method can be used to observe the sizes and distribution of ice crystals, but less finely for the shapes and roundness of the ice crystals. In addition, the samples prepared by the freeze drying method can also be used to observe the damaging of the micro-structures of samples by ice crystals [43].

Paraffin section is a means to observe the changes in tissue structure in samples after freezing and thawing by light microscopy [44]. The preparation process of this method is complicated, including fixation, sampling, dehydration, wax dipping, embedding, sectioning, and dyeing [45]. This method cannot directly observe the characteristics of ice crystals but can infer the role of ice crystals through the damage caused by ice crystals to structures. However, this method is still a common method for food freezing because this method can clearly observe the changes caused by ice crystals to the fine structures of tissues such as the myofascial membrane and endometrium of animal food (Figure 6C) [44].

**Figure 6 foods-13-02773-f006:**
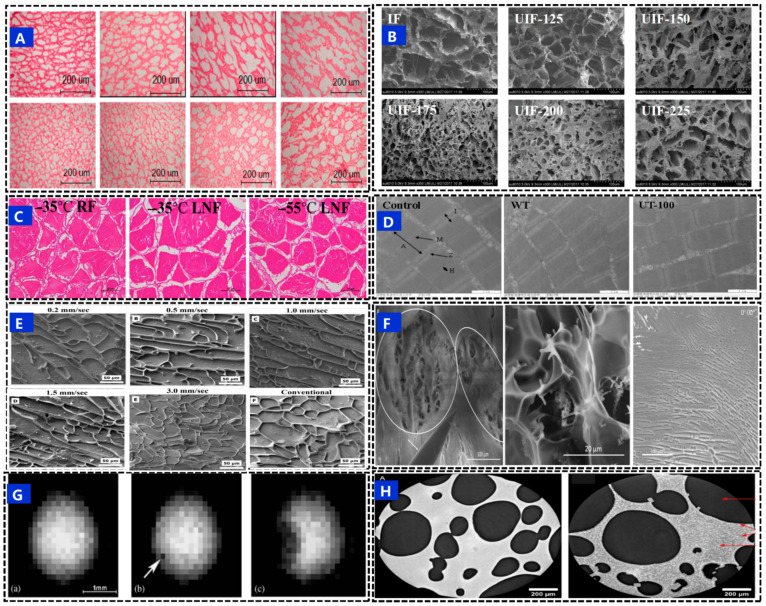
Different methods for characterizing ice crystals: (**A**) light microscopy (frozen section) Reprinted/adapted with permission from Ref. [42] Copyright 6 October 2018, copyright Qinxiu Sun, Fangda Sun, Xiufang Xia, Honghua Xu, Baohua Kong. (**B**) scanning electron microscopy Reprinted with permission from Ref [28]. Copyright 1 July 2019, copyright Qinxiu Sun, Xinxin Zhao, Chao Zhang, Xiufang Xia, Fangda Sun, Baohua Kong. (**C**) light microscopy (paraffin sections) Reprinted with permission from Ref [44]. Copyright 16 April 2022, Zuomiao Yang, Shucheng Liu, Qinxiu Sun, Ouyang Zheng, Shuai Wei, Qiuyu Xia, Hongwu Ji, Chujin Deng, Jiming Hao, Jie Xu. (**D**) transmission electron microscopy. Reprinted with permission from Ref [46]. Copyright 1 April 2021, copyright Qinxiu Sun, Baohua Kong, Shucheng Liu, Ouyang Zheng, Chao Zhang. (**E**) cryo-electron microscopy. Reprinted with permission from Ref [47]. Copyright 1 October 2022, copyright James D. Gillis, William V. Holt, Linda M. Penfold, Kathryn J. Woad, James K. Graham, Julie A. Watts, David S. Gardner, Lisa Yon. (**F**) environmental scanning electron microscopy. Reprinted with permission from Ref [48]. Copyright 30 July 2020, copyright Ľubica Vetráková, Vilém Neděla, Jiří Runštuk, Eva Tihlaříková, Dominik Heger, Evgenyi Shalaev. (**G**) magnetic resonance imaging (^1^H spin-density images of the nucleation and recalescence of a 20% *w*/*w* sucrose droplet freezing in still air at −25 °C. The time between frames was 0.5 s). Reprinted with permission from Ref [49]. Copyright 1 May 2004, copyright J.P. Hindmarsh, C. Buckley, A.B. Russell, X.D. Chen, L.F. Gladden, D.I. Wilson, M.L. Johns. and (**H**) X-ray imaging. Reprinted with permission from Ref [50]. Copyright 1 December 2022, copyright Amira Zennoune, Pierre Latil, Frederic Flin, Jonathan Perrin, Timm Weitkamp, Mario Scheel, Christian Geindreau, Hayat Benkhelifa, Fatou-Toutie Ndoye.

### 5.2. Electron Microscope

Electron microscopy uses an electron beam as the light source and therefore has a higher resolution than optical microscopy (only 200 nm), with a resolution of about 5 nm. Since electron scattering caused by volatile molecules in the sample may cause blurred imaging, the sample should be dried or frozen before observation by conventional electron microscopy. Depending on the imaging method, there are two types of electron microscopy techniques: scanning electron microscopy (SEM) and transmission electron microscopy (TEM) [46].

SEM is a method used to observe the surface microstructures of biological samples. Because it can magnify objects by 10 to 300,000 times [8], SEM can provide fine information that light microscope cannot display. This is because the electrical signal produced by high-energy incoming electrons striking the sample surface is the foundation of SEM imaging. Unlike TEM, SEM does not have tight requirements for sample thickness, and sample preparation is simpler. Before observing the microstructure of a sample by SEM at room temperature, the sample always needs to be fixed, dehydrated, freeze-dried, and sprayed with gold. At the same time, freeze drying can be used instead of fixing, dehydration, and freeze drying to prepare SEM samples for observing the sizes of ice crystals in frozen foods (Figure 6B) [29].

Cryo-electron microscopy (cryo-EM) is a method that won the 2017 Nobel Prize [8]. Compared with normal-temperature SEM, cryo-EM sample preparation is simpler, and it does not need a tedious sample pre-treatment process. The cryo-EM permits the direct observation of frozen materials due to its unique cooling system [51]. Frozen samples can therefore be used right away or unfrozen samples can be placed under a microscope and gradually frozen. Cryo-EM can be used to analyze the interior three-dimensional structures of wet samples at low temperatures by secondary or backscattered electron imaging and can avoid cell contraction because there is no time-consuming pre-treatment. Meanwhile, cryo-EM can also be used to study the nucleation temperatures, sizes, shapes, and positions of ice crystals in food and the mechanical effects of ice crystals.

Unlike other types of SEM, environmental SEM (ESEM) does not require sample drying and conductive pre-treatment but has the advantage of being able to examine virtually any material in its natural state. ESEM can also provide high-resolution images in a saturated water vapor atmosphere while keeping the sample moist. Thus, ESEM is utilized to directly detect both freeze–thawed samples [43] as well as ice crystals in freeze-dried samples [41]. When the details of the sample are required, cryo-EM is superior to ESEM due to its high resolution, but cryo-EM is more expensive and requires a longer elapsed time; therefore, ESEM is more suitable when a wide range of vision is required.

In TEM observation, the sample is penetrated by a short-wavelength electron beam and information on the internal refinement of the sample can be collected. Since the electron beam scatters easily, TEM samples need to be tens of nanometers thick. The pre-treatment of TEM samples is tedious, including sample fixation, polymerization, and staining pre-treatment [35]. TEM is typically used to examine the influence of ice crystals on the interior structures of frozen foods rather than the ice crystals themselves. The ice crystals are characterized indirectly by the degree of disruption of the tissue structure by the ice crystals [43].

### 5.3. DSC

DSC is an accurate method for characterizing the state and content of water [52]. It allows one to program temperature changes to crystallize or melt a sample to study heat absorption or exothermic reactions, allowing one to calculate the amount of freezable and non-freezable water. Sample preparation and transfer to the DSC chamber must be achieved with caution. Longer transfer durations under ambient settings may result in some moisture condensation on the sample pan walls (since the sample container is held at sub-zero temperatures), resulting in an overall rise in pan weight, which might lead to erroneous end results [6].

DSC is frequently employed in conjunction with other techniques because it is a non-in-line measuring method that also damages the material to some extent and only delivers the reaction temperature and associated enthalpy [53]. In addition, DSC is an accepted method for measuring the freezing point, which is the time it takes for water to turn into ice [54]. Another way to use DSC to assess freezing damage in food (mainly meat products) is to compare the enthalpy of denaturation of muscle since freezing treatments may result in protein denaturation due to ice crystal formation and cryoconcentration effects [42].

### 5.4. X-ray

Another new method used to evaluate frozen products is the application of X-rays. X-ray pictures can intuitively represent the internal structures of samples due to the photon energy range of 0.1 to 120 keV and they can be utilized for ice crystal monitoring in frozen meals without sample preparation or chemical fixation [55]. Using a combination of X-ray microscopy and tomography algorithms, X-ray computed tomography can generate 3D (three-dimensional) or 4D (four-dimensional) virtual [56] structures with axial and lateral resolutions up to a few microns of a sample based on X-ray attenuation (absorption and scattering) differences resulting from material density differences within the sample, thereby providing qualitative and quantitative information about the internal structure [57]. Furthermore, X-ray imaging allows for numerous scans of the same sample under varied conditions, which is critical for understanding ice crystal dynamics. X-ray technology is also commonly used to study pores created by ice crystals following freeze drying, where the material can be imaged directly in the frozen state or freeze-dried before imaging. Freeze drying or lyophilization processes sublimate water from the core of the material, leaving a network of gaps and pores that allows imaging at room temperature. Although X-ray imaging is an accurate method to determine the morphology, size distribution, and volume fraction of ice crystals, X-ray imaging does not show the structural details, and freeze drying can lead to the destruction of the cell structure through shrinkage.

### 5.5. Nuclear Magnetic Resonance (NMR) and Magnetic Resonance Imaging (MRI)

Because water has a direct impact on the stability and quality of frozen foods, it is critical to investigate variations in water status in frozen foods. In recent years, NMR and MRI have been developed as non-destructive techniques to monitor water migration and distribution [8]. Freezing can transform a portion of bound water into free water in food tissues due to protein denaturation and water redistribution induced by multiple freeze–thaw cycles of ice crystals, and the mobility of free water increases after the freeze–thaw cycles [8]. As a result, NMR technology can indirectly react to the development and growth of ice crystals. Furthermore, because ice crystallization can reduce the relaxation time T2, NMR is another method for obtaining water information [41].

NMR can provide information on the content, distribution, and mobility of water in frozen foods by reflecting the degree of binding and content changes including free water, bound water, and immobilized water through relaxation time and peak area. However, while NMR cannot generate spatial images, MRI, a more advanced NMR device, can be utilized to generate spatial spectra and relaxation measurement data that clearly depict changes in the distribution state of moisture in frozen foods. MRI has been demonstrated to be beneficial for monitoring ice formation and determining the freezing time during food freezing. The production of ice crystals in this situation is highlighted by a sharp reduction in the intensity of the spatially confined NMR signal. NMR transverse (*T*_2_) relaxation techniques and MRI techniques have also been used to detect dynamic changes in the structures and textures of food products during frozen storage because the distribution of water in food products is altered during freezing/thawing [58].

In a study, the unfrozen liquid mass fraction ratio of the droplets could be determined as a function of post-nucleation time using MRI images and this was then utilized to predict the crystal growth rate throughout the re-aging and following freezing phases. The authors concluded that the MRI technique had proven to be a very valuable tool for analyzing the freezing behavior of small droplets as far as structural and compositional transitions were concerned [59].

## 6. Modeling of Ice Crystal Formation

At present, the understanding of ice crystals is mainly based on experiments, and the research on ice crystal growth laws from the aspects of mechanism and essence is limited. It is important to reveal the process of ice crystal formation and growth law in order to better control the formation of ice crystals. However, there are many factors affecting ice crystal formation, and the formation process of ice crystals is complicated, which will cause some difficulties in studying the formation mechanism of ice crystals. Numerical simulation is not limited by experimental conditions but also can change the conditions at any time and quickly so as to better control variables and make the predictive control of many variables possible [13]. Numerical simulations allow the visualization of the obtained data, which provides the possibility to study the formation mechanism of ice crystals from the microscopic point of view.

At present, numerical simulation techniques are widely used in the simulation study of ice crystal formation. Probabilistic, deterministic, and phase-field methods are currently widely used numerical simulation methods.

Probabilistic methods, or stochastic methods, are based on the core idea of random sampling. A probabilistic model is built on the basis of transforming the problem under study into a stochastic one, random sampling is performed on the model, and subsequently, the sampling results are processed to obtain a numerical solution to the problem. Commonly used probabilistic methods mainly include Monte Carlo [60] and cellular automata methods.

By randomly selecting numbers from the probability distribution of random variables, the Monte Carlo method generates a random numerical sequence that corresponds with the probability distribution characteristics of random variables. When using the Monte Carlo method, the generated random number sequence must comply to the random variable’s unique probability distribution for generating various random number sequences with a specific and uneven probability distribution. The Monte Carlo simulation method can provide a more approximate simulation of the water icing phase transition process and get the best process and probability distribution from the uncertain process of water icing, so many researchers use the Monte Carlo simulation method to study the problem of the water icing process. However, it does not consider the details of macro–micro transport and thus cannot be used to quantify the physical phenomena involved in microstructure growth.

Based on the principle of statistical physics, cellular automata describe the microstructure characteristics and variation laws of materials with mathematical models. Cellular automata differ from Monte Carlo in that they have a relative physical background and certain physical basis. Before simulation, the solidification process is assumed to be stable, the influence of solute concentration change in the solidification process is ignored, and the preferred orientation of dendrite growth is also considered so that the outer profile of dendrite can be displayed. However, the model of grain simulation is single, and the solid–liquid interface must be tracked, and quantitative simulation can not be carried out.

MD simulation is a powerful computer simulation tool for tracking the time evolution of particle systems that interact (such as atoms, molecules, and coarse-grained particles) [61]. The simulation results are consistent with the grain growth state. MD simulations are widely used to study the water phase transition and the structure, dynamics, and thermodynamics of its complexes and solve various problems including ice nucleation [62]. The core idea behind MD simulations is to construct atomic trajectories for a finite system of particles by numerical integration, which numerically integrates a set of Newton’s equations of motion for all particles in the system at a given boundary condition, initial position, and velocity [63]. Furthermore, in order to characterize the interaction of atoms or molecules, it is necessary to specify the potential energy of the particle system. The detailed expression of potential energy and its parameters come from experimental work and advanced quantum mechanics calculation.

To gain a better understanding of the specific events that lead to cell damage caused by extracellular and intracellular ice production during freezing, it is necessary to examine the formation of extracellular and intracellular ice so as to characterize the influence of several core factors such as anisotropy, supercooling, and thermal noise. Among the different methods established, the phase-field method, which has been widely utilized in modeling solidification processes in materials science, is quite versatile in defining crystal formation [61]. Based on Ginzburg–Landau theory, the phase-field method introduces phase-field variables as order parameters to distinguish the solid phase from liquid phase in a supercooled melt. The phase-field method uses consistent governing equations (phase-field, temperature field, and other partial differential equations) and uses a continuous dispersion interface instead of a traditional sharp interface, which effectively avoids tracking a complex solid–liquid interface. It can also be coupled with other physical fields to quantitatively study the influence of various factors on dendrite growth.

Various molecular simulation methods are often combined with experimental methods [including sum-frequency generation spectroscopy [64], Raman spectroscopy, infrared spectroscopy, and X-ray techniques and scanning probe microscopies (Shimizu et al., 2018)] to study structural changes at the ice–water interface and ice–water interactions under different conditions according to practical needs [65].

## 7. Conclusions and Future Perspectives

During the freezing process, water molecules form ordered network structures through hydrogen bonding to form ice crystals. Therefore, the hydrogen bonding interactions between water molecule clusters are important forces that affect the freezing process. Fully understanding the hydrogen bonding interactions between water molecule clusters in different freezing processes is of great significance for revealing and regulating the freezing process. Due to the different pressure and temperature conditions under which ice crystals form, different types of ice crystals can also be formed, determining their absolute density, shapes, and distribution, which in turn affect the impact of ice crystals on frozen foods. In this review, the structural changes in water clusters under different hydrogen bonds and the structural types of ice crystals under different temperatures and pressures were reviewed in order to provide a theoretical basis for revealing the new technical mechanism of controlling the formation and growth of ice crystals. Many new freezing technologies are based on these theories to control the sizes and morphology of ice crystals, thereby improving the quality of frozen food. For example, magnetic-field-assisted freezing technology and electric-field-assisted freezing technology regulate the nucleation of ice crystals by changing the hydrogen bonding between water molecule clusters, the ultrasound-assisted freezing technique induces ice crystal nucleation by generating cavitation bubbles as crystal nuclei, pressure-assisted freezing technology controls the phase transition temperature of water and the structures of ice crystals by changing the pressure, liquid gas freezing technology (liquid nitrogen freezing, liquid CO_2_ freezing technology) and ultrasonic-assisted freezing technology regulate ice crystals by accelerating mass and heat transfer, and the use of antifreeze agents (such as polysaccharides, polypeptides, natural deep eutectic solvents, etc.) regulates the formation of ice crystals by changing the state of water molecules. Although the structures of water molecules and ice crystals have been known to some extent, there is still a lack of in-depth research on the intrinsic relationship between them. Therefore, the mechanism by which many new technologies control the formation of ice crystals is still unclear, and the continuous efforts of researchers are still needed to promote their industrial application.

The development of characterization methods of ice crystals is the basis of people’s research on freezing technology. Although the development of some electron microscopies (such as cryo-electron microscopy) provides the possibility to observe the formation of ice crystals during freezing, these technologies either cannot observe the formation of ice crystals during the application of new freezing technologies in situ or are rarely used because the equipment is too expensive. Scanning electron microscopes or light microscopes are commonly used to observe the sizes and morphology of ice crystals and observe the microstructure of food indirectly, or other indicators are used to indirectly prove the influence of ice crystal formation, such as low-field nuclear magnetic resonance technology, X-rays, nuclear magnetic resonance imaging technology, and some spectral technologies. At present, conventional experimental methods find it difficult to explain the changes in the subtle environment and forces during freezing. Computer simulation technologies such as molecular dynamics simulation and Monte Carlo simulation provide the possibility to clarify the changes in the water molecular structure and ice crystals during freezing, which makes people have a deeper understanding of the formation process of ice crystals.

## Figures and Tables

**Figure 1 foods-13-02773-f001:**
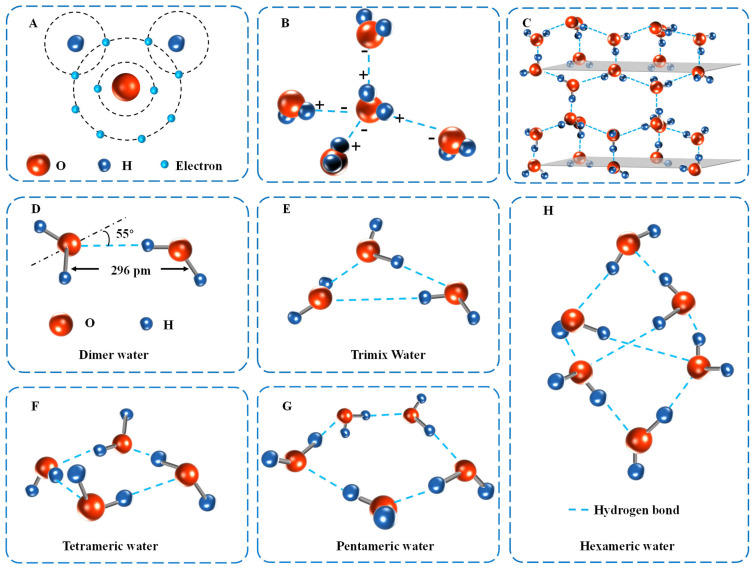
Typical cluster structures of water molecules. (**A**) Electron distribution of water molecules. (**B**) Polarity of water molecules. (**C**) Crystalline structure of water molecules. (**D**) Structure of dimeric water molecular clusters. (**E**) Structure of trimeric water molecular clusters. (**F**) Structure of tetrameric water molecular clusters. (**G**) Structure of pentameric water molecular clusters. (**H**) Structure of hexameric water molecular clusters.

**Figure 2 foods-13-02773-f002:**
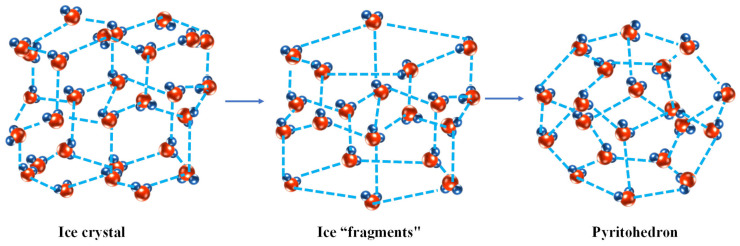
Schematic diagram of forming polyhedron in ice fusion processes.

**Figure 3 foods-13-02773-f003:**
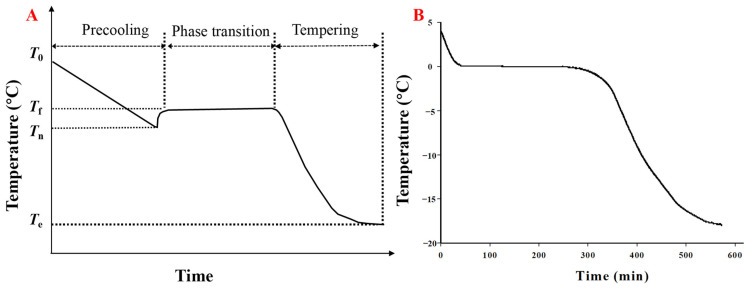
Typical freezing curve of water (**A**) and common carp (**B**). Reprinted with permission from Ref. [28]. Copyright 1 July 2019, copyright Qinxiu Sun, Xinxin Zhao, Chao Zhang, Xiufang Xia, Fangda Sun, Baohua Kong. Abbreviations: *T*_0_, initial temperature; *T*_f_, freezing temperature; *T*_n_, nucleation temperature; *T*_e_, freezing endpoint temperature.

**Figure 4 foods-13-02773-f004:**
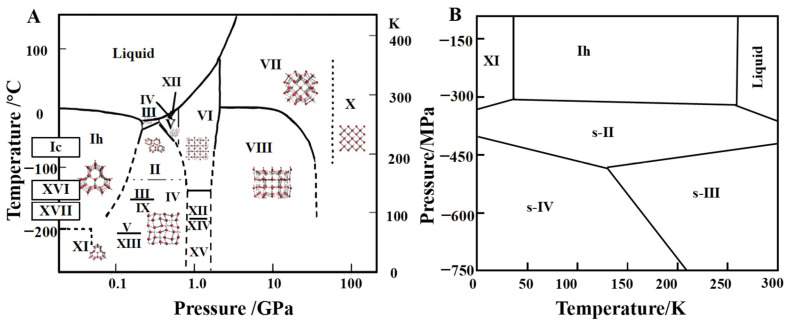
(**A**) Solid–liquid phase diagram of ice under positive pressure. Reprinted with permission from Ref. [32]. Copyright 24 May 2012, copyright Thorsten Bartels-Rausch et al. (**B**) Solid–liquid phase diagram of ice under negative pressures with the TIP4P/2005 model.

**Table 1 foods-13-02773-t001:** Characterization methods for ice crystals.

Method		Sample Processing	Observation Content	Summary of Characteristics
Light microscopy	Frozen section	Freeze substitution or freeze drying	Fractal dimension, ice crystal diameter, ice crystal roundness, elongation, ice crystal area.	Indirectly observing the sizes, shapes, and distribution of ice crystals by observing the surface of the sample
	Paraffin section	Samples often need to be fixed, dehydrated, waxed, embedded, sliced, and stained	Tissue structure of samples	Indirectly characterizing the effect of ice crystals on food
Electron microscope	Scanning electron microscopy (SEM)	Sample always needs to be fixed, dehydrated, freeze-dried, and sprayed with gold. Freeze drying can be used instead of fixing, dehydration, and freeze drying	Surface microstructure of samples, or the size of ice crystals in frozen food	SEM can provide fine information that light microscope cannot display. Unlike TEM, SEM does not have tight requirements for sample thickness, and sample preparation is simpler.
	Cryo-electron microscopy	No need for sample preparation	Interior three-dimensional structure of wet samples at low temperature or the nucleation temperature, sizes, shapes, and positions of ice crystals in food and the mechanical effects of ice crystals.	
	Environmental SEM (ESEM)	No need for sample preparation and conductive pre-treatment	Ice crystals in freeze-dried samples	
	Transmission electron microscopy (TEM)	Pre-treatment of TEM samples is tedious, including sample fixation, polymerization, and staining pre-treatment	Internal structure of frozen food	The sample preparation process is complex, indirectly reflecting the role of ice crystals by collecting refined information within the sample
DSC		No need for sample preparation	Amount of freezable and non-freezable water, freezing point, and associated enthalpy	Indirectly reflecting the impact of ice crystals on food through changes in freezing parameters
X-ray		No need for sample preparation. It can be imaged directly in a frozen state or freeze-dried before imaging	Used for monitoring ice crystals in frozen foods or studying the pores formed by ice crystals after freeze drying	Although X-ray imaging is an accurate method to determine the morphology, size distribution, and volume fraction of ice crystals, it does not show the structural details, and freeze drying can lead to the destruction of cell structure through shrinkage.
Nuclear magnetic resonance		No need for sample preparation	The state of water, including bound water, free water, and immobilized water	Indirectly characterizing the effect of ice crystals on food structure through changes in moisture state
Magnetic resonance imaging		No need for sample preparation	The spatial distribution status of water	

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
