# Peer review of "Basic Theory of Ice Crystallization Based on Water Molecular Structure and Ice Structure"

_foods, 2024, doi:10.3390/foods13172773_

Round 1
Reviewer 1 Report
Comments and Suggestions for Authors
In the present study the authors claim that this review provides more information on ice crystallization and promotes the control of ice crystals in frozen foods. After reading the document I have doubts as to whether they have achieed this objective. Indeed, they provide descriptive information on ice crystallization, but I find that there is a lack of a section on recommendations or future trends that suggests in which direction progress should be made to control ice crystals in frozen foods.
In my opinion there is some conceptual error. For example in fig. 3 presents the Freezing Curve of water or a simple component. Foods are much more complex and the phases indicated are not as clear. The authors should review the literature and provide a figure more in line with the freezing curve in foods.
Between lin. 193-206 the information is duplicated. Please revise.
In lin. 227 it is stated "Each increase in super-cooling increases the rate of ice nucleation approximately 10 fold". This statement is very generic, they should be more precise. E.g. "every XºC in super-cooling increases....."
Author Response
Q1: In the present study the authors claim that this review provides more information on ice crystallization and promotes the control of ice crystals in frozen foods. After reading the document I have doubts as to whether they have achieed this objective. Indeed, they provide descriptive information on ice crystallization, but I find that there is a lack of a section on recommendations or future trends that suggests in which direction progress should be made to control ice crystals in frozen foods.
A1: Thank you very much for your suggestion. We have added the future trends to the manuscript as follows:
- Conclusions and future perspectives
Freezing is currently the most common method of food preservation, but the mechanical extrusion of ice crystals formed by water and the concentration effect of solvents inevitably deteriorate the quality of frozen food. The size, morphology and distribution of ice crystals determine the quality of frozen food, which depends on the nucleation and growth of ice crystals. The nucleation and growth of ice crystals are controlled by the structure of water molecular clusters and ultimately affect the structure of ice crystals after formation. In this review, the structural changes of water clusters under different hydrogen bonds and the structural types of ice crystals under different temperatures and pressures are reviewed. In order to provide a theoretical basis for revealing the new technical mechanism of controlling the formation and growth of ice crystals. Many new freezing technologies are based on these theories to control the size and morphology of ice crystals, thereby improving the quality of frozen food. For example, magnetic field-assisted freezing technology and electric field-assisted freezing technology that regulate the nucleation of ice crystals by changing the hydrogen bonding between water molecule clusters; Ultrasound-assisted freezing technique for inducing ice crystal nucleation by generating cavitation bubbles as crystal nuclei; Pressure-assisted freezing technology that controls the phase transition temperature of water and the structure of ice crystals by changing the pressure; Liquid gas freezing technology (liquid nitrogen freezing, liquid CO2 freezing technology) and ultrasonic-assisted freezing technology that regulate ice crystals by accelerating mass and heat transfer; The use of antifreeze agents (such as polysaccharides, polypeptides, natural deep eutectic solvents, etc.) that regulate the formation of ice crystals by changing the state of water molecules. Although the structures of water molecules and ice crystals have been known to some extent, there is still a lack of in-depth research on the intrinsic relationship between them. Therefore, the mechanism by which many new technologies control the formation of ice crystals is still unclear, and the continuous efforts of researchers are still needed to promote their industrial application.
Q2: In my opinion there is some conceptual error. For example in fig. 3 presents the Freezing Curve of water or a simple component. Foods are much more complex and the phases indicated are not as clear. The authors should review the literature and provide a figure more in line with the freezing curve in foods.
A2: Thank you very much for your suggestion. The typical freezing curve of complex foods (such as fish) has been added to Figure 3, and the relevant discussion has also been revised in the manuscript.
Fig. 3 Typical freezing curve of water (A) and common carp (B). T0, Initial temperature; Tf, Freezing temperature; Tn, Nucleation temperature; Te, Freezing endpoint temperature.
Q3: Between lin. 193-206 the information is duplicated. Please revise.
A3: Thank you very much for your suggestion. We have corrected the relevant errors.
Q4: In lin. 227 it is stated "Each increase in super-cooling increases the rate of ice nucleation approximately 10 fold". This statement is very generic, they should be more precise. E.g. "every XºC in super-cooling increases....."
A4: Thank you for your suggestion. This sentence has been advised to “According to the classical theory of water freezing nucleation, for every 1 °C decrease in supercooling temperature, the nucleation rate increases geometrically [24].”.
Newly cited reference:
- Ickes, L., André W., Hoose, C., & Lohmann, U. Classical nucleation theory of homogeneous freezing of water: thermodynamic and kinetic parameters. Phys Chem Chem Phys 2015, 17(8), 5514-5537.

Reviewer 2 Report
Comments and Suggestions for Authors
This review systematically examines the ice crystallization process, including the ice crystal formation mechanism, water molecules and crystal structure, and crystal characterization methods. The review is thorough and informative; however, some alterations are required. The review could benefit from adding more practical examples that illustrate the application of the discussed theories and methods.
Specific comments:
Page 1, lines 39-41, please, briefly elaborate on each factor that determines the degree of mechanical damage to food tissue.
Pages 1-2, line 48, please, consider including, in the Introduction section, a more detailed discussion of the practical implications of ice crystallization in different types of food products.
Page 3,, lines 124-127, please provide reference for the “window effect”.
Page 4, lines 128-133, please provide a reference for the method for obtaining the polymerized water.
Page 5, lines 193-207, this section is from the Foods template explanations about Material and Methods, please, remove it.
Page 6, lines 223-224, please elaborate more about factors affecting super-cooling (the nature of the solution, the cooling rate, the process conditions, and the roughness of the vessel surface).
Page 10, in the section Characterization method of ice crystals, please consider including a Table that summarizes characterization methods before elaborating on them.
On page 15, lines 616-644, please consider rewriting the Conclusion section to make it more concise and include the key points of the review.
Referencing through the review should be changed, with the reference numbers placed in square brackets [ ], and placed before the punctuation.
Author Response
This review systematically examines the ice crystallization process, including the ice crystal formation mechanism, water molecules and crystal structure, and crystal characterization methods. The review is thorough and informative; however, some alterations are required. The review could benefit from adding more practical examples that illustrate the application of the discussed theories and methods.
Specific comments:
Q1: Page 1, lines 39-41, please, briefly elaborate on each factor that determines the degree of mechanical damage to food tissue.
A1: Thank you very much for your suggestion. The following factors have been added to the manuscript:
The size, shape, and distribution of ice crystals during freezing determine the degree of mechanical damage to food tissue, which depends on factors such asnucleation temperature, freezing point, temperature fluctuations, moisture state, super-cooling, and freezing rate (Tan, Mei, & Xie, 2021).
Newly cited reference:
- Tan, M., Mei, J., & Xie, J.The formation and control of ice crystal and its impact on the quality of frozen aquatic products: a review.Crystals 2021, 11(1), 68.
Q2: Pages 1-2, line 48, please, consider including, in the Introduction section, a more detailed discussion of the practical implications of ice crystallization in different types of food products.
A2: Thank you very much for your suggestion. The relevant content as follow has been added to the manuscript.
In frozen tissue foods, the formation of large ice crystals outside the cell leads to various harmful effects on the tissue, including mechanical damage, low-temperature concentration, frozen burns, and recrystallization. These phenomena may occur in metabolic system disorders, enzyme system misalignment, cell membrane damage, and ultimately lead to cell lysis. On the other hand, the formation of uniformly distributed fine crystals inside and outside the cell can better preserve the quality of the product to the tissue due to minimal damage (Dalvi-Isfahan, Jha, Tavakoli, Daraei-Garmakhany, Xanthakis, & Le-Bail, 2019). For ice cream products, the quantity and size of ice crystals directly determine whether the taste of ice cream is smooth or rough. Large ice crystals can disrupt the smooth texture of ice cream. Not only that, ice crystals also determine the skeleton of ice cream. However, crystals are more needed in processes such as freeze-drying and freeze concentration (Kiani, & Sun, 2011). Therefore, controlling, understanding, and predicting the ice crystallization process and related phenomena is crucial for improving food freezing technology.
Newly cited references:
- Dalvi-Isfahan, M., Jha, P. K., Tavakoli, J., Daraei-Garmakhany, A., Xanthakis, E., & Le-Bail, A. Review on identification, underlying mechanisms and evaluation of freezing damage. J. Food Eng.2019, 255, 50-60.
- Kiani, H., & Sun, D. W. Water crystallization and its importance to freezing of foods: a review. Trends Food Sci.Tech. 2011, 22(8), 407-426.
Q3: Page 3, lines 124-127, please provide reference for the “window effect”.
A3: Thank you very much for your suggestion. Sorry, Our definition of the window effect is not accurate. Dimeric water can absorb infrared radiation, causing a decrease in temperature. We have added relevant reference to the manuscript.
Newly cited reference:
- Suck, S. H., Wetmore, A. E., Chen, T. S., & Kassner, J. L. Role of various water clusters in IRabsorption in the 8-14-m window region. A.O. 1982, 21(9), 1610-1614.
Q4: Page 4, lines 128-133, please provide a reference for the method for obtaining the polymerized water.
A4: Thank you very much for your suggestion. The relevant reference has been added to the manuscript.
Newly cited reference:
- Michalarias, I., Gao, X. L., Ford, R. C., & Li, J. C. Recent progress on our understanding of water around biomolecules. J. Mol. Liq.2005, 117(1-3), 107-116.
Q5: Page 5, lines 193-207, this section is from the Foods template explanations about Material and Methods, please, remove it.
A5: Thank you for your suggestion. This section has been removed.
Q6: Page 6, lines 223-224, please elaborate more about factors affecting super-cooling (the nature of the solution, the cooling rate, the process conditions, and the roughness of the vessel surface).
A6: Thank you very much for your suggestion. The following contents have been added to the manuscript:
It is related to the nature of the solution, the cooling rate, the process conditions, and the roughness of the vessel surface. The surfaces, suspended particles, impurities, and tiny bubbles of heat exchangers and containers can promote heterogeneous nucleation and reduce supercooling. For pure supercooled liquids, they need to undergo spontaneous density or composition fluctuations, undergo homogeneous nucleation, and the degree of supercooling is related to the density of the solution. The degree of supercooling also depends on the cooling rate. However, there is no unified conclusion on the relationship between supercooling and cooling rate. Some researchers believe that increasing the cooling rate can increase supercooling, while others believe that increasing the cooling rate can reduce supercooling. In addition, researchers have also found that the higher the surface roughness of the container, the lower the supercooling (Safari et al., 2017).
Newly cited reference:
- Safari, A., Saidur, R., Sulaiman, F. A., Xu, Y., & Dong, J. A review on supercooling of phase change materials in thermal energy storage systems. Rene.Sust.Ene. Rev. 2017, 70, 905-919.
Q7: Page 10, in the section Characterization method of ice crystals, please consider including a Table that summarizes characterization methods before elaborating on them.
A7: Thank you very much for your suggestion. A table summarizing the characterization methods of ice crystals has been added to the manuscript.
Table 1 Characterization methods of ice crystals
Method |
Sample processing |
Observation content |
Summary of characteristics |
|
Light microscopy |
Frozen section |
Freeze substitution or freeze drying |
Fractal dimension, ice crystal diameter, ice crystal roundness, elongation, ice crystal area. |
Indirectly observe the size, shape, and distribution of ice crystals by observing the surface of the sample |
Paraffin section |
Samples often need to be fixed, dehydrated, waxed, embedded, sliced, and stained |
Tissue structure of samples |
Indirectly characterizing the effect of ice crystals on food |
|
Electron microscope |
Scanning electron microscope (SEM) |
Sample always needs to be fixed, dehydrated, freeze-dried and sprayed with gold. Freeze-drying can be used instead of fixing, dehydration and freeze-drying |
Surface microstructure of samples, or the size of ice crystals in frozen food |
SEM can provide fine information that light microscope cannot display. Unlike TEM, SEM does not have tight requirements for sample thickness, and sample preparation is simpler. |
Cryo-electron microscopy |
No need for sample preparation |
Interior three-dimensional structure of wet samples at low temperature, or the nucleation temperature, size, shape and position of ice crystals in food, and the mechanical effect of ice crystals. |
||
Environmental SEM (ESEM) |
No need for sample preparation and conductive pretreatment |
Ice crystals in freeze-dried samples |
||
Transmission electron microscopy (TEM) |
Pre-treatment of TEM samples is tedious, including sample fixation, polymerization, and staining pre-treatment |
Internal structure of frozen food |
The sample preparation process is complex. Indirectly reflecting the role of ice crystals by collecting refined information within the sample |
|
DSC |
No need for sample preparation |
Amount of freezable and non-freezable water, freezing point, and associated enthalpy |
Indirectly reflecting the impact of ice crystals on food through changes in freezing parameters |
|
X-ray |
No need for sample preparation. It can be imaged directly in a frozen state or freeze-dried before imaging |
Used for monitoring ice crystals in frozen foods or studying the pores formed by ice crystals after freeze-drying |
Although X-ray imaging is an accurate method to determine the morphology, size distribution and volume fraction of ice crystals, it does not show the structural details, and freeze-drying can lead to the destruction of cell structure through shrinkage. |
|
Nuclear magnetic resonance |
No need for sample preparation |
The state of water, including bound water, free water, and immobilized water |
Indirectly characterizing the effect of ice crystals on food structure through changes in moisture state |
|
Magnetic resonance imaging |
No need for sample preparation |
The spatial distribution status of water |
Q8: On page 15, lines 616-644, please consider rewriting the Conclusion section to make it more concise and include the key points of the review.
A8: Thank you very much for your suggestion. Combining suggestions from other reviewers, the relevant content has been modified as follows:
- Conclusions and future perspectives
During the freezing process, water molecules form ordered network structures through hydrogen bonding to form ice crystals. Therefore, the hydrogen bonding interactions between water molecule clusters are important forces that affect the freezing process. Fully understanding the hydrogen bonding interactions between water molecule clusters in different freezing processes is of great significance for revealing and regulating the freezing process. Due to the different pressure and temperature conditions under which ice crystals form, different types of ice crystals can also be formed, determining their absolute density, shape, and distribution, which in turn affect the impact of ice crystals on frozen foods. In this review, the structural changes of water clusters under different hydrogen bonds and the structural types of ice crystals under different temperatures and pressures are reviewed. In order to provide a theoretical basis for revealing the new technical mechanism of controlling the formation and growth of ice crystals. Many new freezing technologies are based on these theories to control the size and morphology of ice crystals, thereby improving the quality of frozen food. For example, magnetic field-assisted freezing technology and electric field-assisted freezing technology that regulate the nucleation of ice crystals by changing the hydrogen bonding between water molecule clusters; Ultrasound-assisted freezing technique for inducing ice crystal nucleation by generating cavitation bubbles as crystal nuclei; Pressure-assisted freezing technology that controls the phase transition temperature of water and the structure of ice crystals by changing the pressure; Liquid gas freezing technology (liquid nitrogen freezing, liquid CO2 freezing technology) and ultrasonic-assisted freezing technology that regulate ice crystals by accelerating mass and heat transfer; The use of antifreeze agents (such as polysaccharides, polypeptides, natural deep eutectic solvents, etc.) that regulate the formation of ice crystals by changing the state of water molecules. Although the structures of water molecules and ice crystals have been known to some extent, there is still a lack of in-depth research on the intrinsic relationship between them. Therefore, the mechanism by which many new technologies control the formation of ice crystals is still unclear, and the continuous efforts of researchers are still needed to promote their industrial application.
Q9: Referencing through the review should be changed, with the reference numbers placed in square brackets [ ], and placed before the punctuation.
A9: Thank you very much for your suggestion. The references have been formatted according to the requirements.

Round 2
Reviewer 1 Report
Comments and Suggestions for Authors
The changes made have improved the document.